# Identifying and prioritising barriers to injury care in Northern Malawi, results of a multifacility multidisciplinary health facility staff survey

John Whitaker[1,2,3]*, Taniel Njawala[4], Vitumbeku Nyirenda[4], Abena S. Amoah[4,5,6], Albert Dube[4], Lindani Chirwa[7,8], Boston Munthali[9,10], Rory Rickard[3], Andrew J. M. Leather[2‡], Justine Davies[1,11,12‡]

1 Institute of Applied Health Research, University of Birmingham, Birmingham, United Kingdom, 2 School of Life Course and Population Sciences, King's College London, London, United Kingdom, 3 Academic Department of Military Surgery and Trauma, Royal Centre for Defence Medicine, Birmingham, United Kingdom, 4 Malawi Epidemiology and Intervention Research Unit (formerly Karonga Prevention Study), Chilumba, Malawi, 5 Faculty of Epidemiology and Population Health, London School of Hygiene & Tropical Medicine, London, United Kingdom, 6 Department of Parasitology, Leiden University Center for Infectious Diseases, Leiden University Medical Center, Leiden, the Netherlands, 7 Karonga District Hospital, Karonga District Health Office, Karonga, Malawi, 8 School of Medicine & Oral Health, Department of Pathology, Kamuzu University of Health Sciences (KUHeS), Blantyre, Malawi, 9 Mzuzu Central Hospital, Department of Orthopaedic Surgery, Mzuzu, Malawi, 10 Lilongwe Institute of Orthopaedic and Neurosurgery, Lilongwe, Malawi, 11 Medical Research Council/Wits University Rural Public Health and Health Transitions Research Unit, Faculty of Health Sciences, School of Public Health, University of the Witwatersrand, Johannesburg, South Africa, 12 Department of Global Surgery, Stellenbosch University, Stellenbosch, South Africa

‡ AJML and JD are joint senior authors on this work.
* j.whitaker@bham.ac.uk

**Data Availability Statement:** Data cannot be shared openly due to risks of identifying healthcare workers. However, data can be made available

## Abstract

### Introduction

The burden of injuries globally and in Malawi is substantial. Optimising both access to, and quality of, care in health systems requires attention. We aimed to establish how health facility staff in Karonga, Malawi, perceive barriers to seeking (delay 1), reaching (delay 2) and receiving (delay 3) injury care.

### Method

We conducted a cross-sectional survey of health facility staff who treat patients with injuries in all health facilities serving the Karonga Demographic Surveillance Site population. The primary outcome was participant perceptions of the importance of delays 1 to 3 following injury. Secondary outcomes were the barriers within each of these delays considered most important and which were considered the most important across all delays stages.

### Results

228 staff completed the survey: 36.8% (84/228) were female and 61.4% (140/228) reported being involved in caring for an injured person at least weekly. Delay 3 was most frequently

from Malawi Epidemiology and Intervention Research Unit (MEIRU) on reasonable request and following the signing of a data transfer agreement. Data enquiries should be sent to info@meiru.mw and should indicate the title of the paper.

**Funding:** "This work was supported by a research fellowship awarded to JW from the Royal College of Surgeons of England and The King's Centre for Global Health and Health Partnerships. The funders had no role in study design, data collection and analysis, decision to publish, or preparation of the manuscript".

**Competing interests:** The authors have declared that no competing interests exist.

considered the most important delay 35.1% (80/228), with 19.3% (44/228) and 16.6% (38/228) reporting delays 1 and 2 as the most important respectively; 28.9% (66/228) of respondents either did not know or answer. For delay 1 the barrier, *"the perceived financial costs associated with seeking care are too great"*, was considered most important. For delay 2, the barrier *"lack of timely affordable emergency transport (formal or informal)"* was considered most important. For delay 3, the barrier, *"lack of reliably available necessary physical resources (infrastructure, equipment and consumable material)"* was considered most important. When considering the most important overall barrier across all delays, the delay 3 barrier, *"lack of reliably available necessary physical resources"* received the most nominations (41.7% [95/228]).

## Conclusions

Given the awareness of health facility staff of the issues facing their patients, these findings should assist in informing health system planning.

## Introduction

Injuries represent a substantial burden of disease, accounting for 8% of deaths worldwide [1]. Low and Middle Income Countries (LMICs) account for 90% of these injury related deaths [2]. Non-fatal injuries are similarly substantial with a billion people sustaining injuries significant enough to need care each year [3]. A third of injury related deaths could be avoided were LMICs able to achieve similar case fatality rates seen in high income countries [4]. Greater research and investment into health systems in these settings is therefore justified. Improvements in both access to and quality of injury care require attention [5].

Within Malawi, injury represents 6% of all deaths and one fifth of non-communicable disease and injury (NCDI) disease adjusted life years (DALYs) [6]. Economically productive younger members of the population are disproportionately affected; two thirds of Malawians are aged under 40 [7] but this age-group experience 82% of injury burden [6]. The health system is deficient in facility-based human and physical resources required for injury care [8].

Attempts to assess and strengthen injury care systems should be holistic in their approach [9]. However, most research into injury care systems has been facility centric with few studies incorporating the wider care ecosystem [10]. The Three Delays framework offers a holistic approach to considering health system factors delaying care seeking (delay 1), reaching a place of care (delay 2) and receiving appropriate, high quality care (delay 3) [11]. Whilst the Three Delays framework was originally described and extensively used for studying maternal and child health [11, 12], it has gained popularity in surgical and injury care research [10, 13–16].

Barriers to urgent care in LMICs can be diverse, multifaceted, and intersecting [17, 18]. Prioritising which barriers to investigate and target for intervention is important. Given health facility staff are well-placed to understand the needs of their patients and to advocate for change, understanding health facility staff perceptions of delays to access to care is important. An earlier Delphi study developed expert consensus on which barriers to injury care are important to assess when evaluating an LMIC injury care health system [14]. We used these Delphi expert consensus findings as a basis from which to understand how local health facility staff perceive and prioritise barriers to trauma care within the Three Delays framework in the setting of Northern Malawi.

## Methods

We conducted a cross-sectional survey of health facility staff in health facilities receiving injured patients in Karonga, Northern Malawi. We have reported this study according to the STROBE checklist [19].

### Study setting

The study setting has been described before [20, 21]. It centred upon the health system accessible to people living in the Karonga Demographic Surveillance Site catchment, Karonga District, Northern Malawi [22]. As a rural lakeshore district of over 350,000, farming and fishing are dominant economic activities [22]. One main metalled road crosses the district with secondary roads mostly unpaved. There are primary facilities run by the government (including a military facility accessible by civilians), private and faith-based providers; secondary care facilities include one run by the government 70km to the North and a faith-based provider run facility 40km to the South. A tertiary care government run referral facility is based in the regional capital Mzuzu, 150km to the South. Primary facilities are usually staffed by non-physician Medical Assistants and Clinical Officers. No pre-hospital emergency medical service is available [20, 23]. Injured patients commonly present to primary facilities initially rather than bypassing to secondary or tertiary care [20, 23]. Whilst those with secondary education can speak English, other facility staff speak the vernacular language of Tumbuka.

### Data collection

**Survey development.**   The survey was constructed according to the Three Delays framework and developed using barriers elicited from the Delphi exercise [14]. The survey was translated from English to Tumbuka and back translated to confirm accuracy and retention of meaning. The survey was piloted amongst four native Tumbuka speaking healthcare workers not working at the included facilities and minor adjustments made to improve comprehension.

**Identification of participants.**   All eleven facilities serving the Karonga Demographic Surveillance Site catchment (8 primary facilities, 2 secondary facilities and one tertiary facility) were visited with the permission of the senior clinical authority at each location. All available staff members during the time of facility visits (e.g. not on leave) who had been involved in an injured person's care in the preceding twelve months were eligible and approached to take part in the survey. Eligible participants included those staff groups who provided direct care and those indirectly involved in injury care provision, such as technical and administrative staff, who might share insight into barriers and delays experienced by injured persons [24].

**Survey conduct.**   The survey was administered to health facility staff between 22[nd] July and 30[th] October 2019 in Tumbuka by two native speaking authors (TN and VN, both qualified Medical Assistants) at a convenient time and quiet location within the facility. Survey responses were collected onto electronic tablets using the REDCap Mobile App and uploaded onto a REDCap [25] server database at the end of each facility visit. Each survey took between 30–60 minutes to complete.

**Survey content.**   Data collected included sex of respondent, frequency of caring for an injured person (mutually exclusive categories of daily, weekly, monthly, quarterly and annually), the name of their place of work, job role (categorised as doctor, nurse, clinical officer, prehospital worker, medical assistant or other, which was then specified), level of training in injury care (mutually exclusive categories of no formal training in the care of the injured, training received during primary healthcare qualification, post-qualification training through a course up to less than ten days in total, or significant post-qualification training including

formal postgraduate qualifications, placements, fellowships or courses of more than ten days) and time since last injury training (mutually exclusive categories of less than one year ago, between 1 and 3 years ago and more than 3 years ago).

Regarding the perceived importance of delay stages, participants were first asked to estimate which of delays 1–3 affected the largest and smallest number of people, caused the most and least delays to those affected, were the easiest and most difficult to change to improve injury care, and overall, which were the most and least important in the health system. Participants were then asked to estimate how many patients (mutually exclusive categories of almost all patients (81–100%), more than half but not all patients (61–80%), about half (41–60%), some, but less than half (21–40%), few (1–20%) and none) experienced any delay seeking care, delay reaching care (with subgroups of reaching care within 24, 12, 6, 2 and 1 hours), or any delay receiving care. Participants were asked to estimate overall time delays incurred in seeking and receiving injury care (mutually exclusive categories of <1 hour, >1 but <2 hours, >2 but <4 hours, >4 but <6 hours, >6 but <12 hours, >12 but <24 hours and >24 hours). Finally, participants were also asked to estimate how much harm came to patients who had experienced a) delays seeking care, b) delays reaching care once a decision to seek care had been made (of between 1–4 hours and more than 4 hours), and c) delays receiving care. Harms were asked as mutually exclusive categories of no or minimal harm, minor harm, slightly prolonged pain or suffering but very low risk of long-term consequences, significant harm that poses a small or moderate risk to causing long term harm or a small risk to life and very significant harm, likely to cause high risk of long-term harm or high risk to life.

Regarding the perceived prominence of barriers within delay stages, participants were then presented with the Delphi [14] derived theoretical barriers to care for each delay stage (S1 Table in S2 File). For each of delays stage, participants were asked to put the barriers in order (from most to least) according to which a) affected the largest number and b) caused the longest amount of delay to those affected. Participants could also propose additional barriers using free text and indicate where they would place them in the above order. Additionally, participants were also asked to select the three barriers from across any of delays 1 to 3 they considered to be the most important.

## Outcome variables

The primary outcomes were participant perceptions of the importance of delays 1 to 3 (estimates of the number of injured people affected at each delay stage, length of delays, and harm associated with delays 1 to 3 and the overall most important delay).

Secondary outcomes were the mean scores for barriers within each of delays 1 to 3, and the number of participants reporting a barrier as being amongst the three most important from across any of delays 1 to 3. Mean barrier scores were calculated as follows. Barriers in each delay ordered by participants were assigned a score corresponding to the order they were placed in that delay. Barriers considered to affect the largest number of people were scored 1, those affecting the second largest number of people were scored 2 etc. A mean score across all participants was then calculated for each barrier for a) affecting the largest number of patients, b) causing the longest amount of delay, and c) a combined mean barrier score (a+b). Lower scores thereby indicate a barrier being perceived as more important within its corresponding delay.

For other descriptive variables, facility types were categorised as primary care or referral (secondary or tertiary care). Frequency of providing injury care was dichotomised to at least weekly and less than weekly, and injury care training level dichotomised into any postgraduate injury care training, including short courses and no postgraduate training. Job role was

dichotomised to clinical care provider (including Doctors, Nurses, Clinical Officers, Medical Assistants and Ambulance care workers) and "other".

## Analysis

Participant characteristics, perceptions of aspects of importance of delays 1 to 3, estimates of the number affected at each delay stage, length of delays and harm associated with delays 1 to 3 were all described with counts and percentages. Barrier scores are described as mean scores. Counts describe each barrier reported within the top 3 most important overall. One author (JW) reviewed free text responses proposing novel barriers and judged whether the proposed barriers were indeed novel or could be subsumed into an existing category.

## Ethics statement

Study participants were provided with a participant information leaflet and consent form which they read or had read out to them in English or Tumbuka. Any questions about the study conduct were answered, and participants signed confirming their written consent to participate in the study. This was witnessed. The study was approved by the Malawi National Health Sciences Research Committee (ref 19/03/2263) and the UK MOD Research and Ethics Committee (ref 960/MODEC/19).

## Results

All eleven facilities approached gave permission for staff to take part. Surveys were completed by 228 health facility staff, of which 36.8% (84/228) were female, 61.4% (140/228) reported being involved in the care of an injured person at least weekly, and 54.4% (124/228) were based at a referral facility (Table 1). Most participants, 53.5% (122/228), were clinical care providers. They included nurses, 26.8% (61/228), clinicians (doctors, clinical officers, or medical assistants), 25.4% (58/228), and ambulance care providers, 1.8% (4/228). The remaining 46.5% (106/228) comprised of other staff, of whom ward or hospital attendants were the largest group 39.2% (40/102).

Overall perceptions of the Three Delays by health facility staff are presented in Fig 1 and show 43.9% (100/228) believed delay 2 affected the largest number of injured people and 44.2% (101/228) believed delay 2 caused the most delay to those injured people. Delay 3 was perceived to be the easiest to change to improve injury care by 57.0% (130/228). Delay 3 was most commonly considered the most important delay 35.1% (80/228), 19.3% (44/228) and 16.6% (38/228) thought delays 1 and 2 were the most important respectively, although 28.9% (66/228) of respondents either did not know or answer.

Regarding delay 1, 52.2% (119/228) of participants estimated that at least half of injured patients have delayed seeking care (Fig 2). Most participants, 61.8% (141/228), believed these delays in seeking care were typically greater than 2 hours (Fig 3). Most participants, 86.4% (197/228), believed that those patients who have delayed seeking care are likely to have suffered at least significant harm because of the delay (Fig 4).

Regarding delay 2, the majority of participants thought "few" or no injured patients reached care within 1 hour (86.4%, 197/228) or 2 hours (57.9%, 132/228) of injury after deciding to seek care (Fig 5). About half of participants, 49.5% (103/228), believed taking more than 1 hour to reach care would result in at least significant harm. This increased to 88.6% (202/228) for those taking more than 4 hours.

Considering delay 3, the majority of participants, 71.5% (163/228), believed "few" or no patients, experienced delays in receiving care at a facility. Similarly, the majority, 57.0% (130/228), believed that typical delays to receiving care were less than 1 hour. Most participants,

**Table 1. Health facility staff survey participant characteristics.**

| | Number (%) N = 228 |
|---|---|
| Sex of health facility staff | |
| Male | 144 (63.2) |
| Female | 84 (36.8) |
| How often do you care for an injured person? | |
| Daily | 76 (33.3) |
| Weekly | 64 (28.1) |
| Monthly | 67 (29.4) |
| Quarterly | 19 (8.3) |
| Annually | 2 (0.9) |
| What is your place of work? | |
| Primary facility | 104 (45.6) |
| Referral facility | 124 (54.4) |
| What is your job role? | |
| Doctor | 8 (3.5) |
| Nurse | 62 (27.2) |
| Clinical Officer | 36 (15.8) |
| Prehospital / Paramedic / Ambulance care worker | 4 (1.8) |
| Medical Assistant | 16 (7.0) |
| Other health facility staff * | 102 (44.7) |

| Which statement best describes your training in the care of injured patients? | Clinical care provider (n = 126) | Other (n = 102) |
|---|---|---|
| No training | 2 (1.6) | 72 (70.6) |
| Training during primary qualification | 102 (80.9) | 30 (29.4) |
| Post qualification training through course <10 days | 19 (15.1) | 0 |
| Post qualification training > 10 days | 3 (2.4) | 0 |
| For those who had received training, when did you last receive it? | Clinical care provider n = 124 | Other (n = 30) |
| Less than one year ago | 19 (15.3) | 0 |
| Between 1 and 3 years ago | 44 (35.4) | 8 (26.7) |
| More than three years ago | 61 (49.2) | 22 (73.3) |

*Breakdown of other health facility staff categories available in S2 Table of S2 File.

67.1% (153/228), believed that those who have experienced delays in receiving care were likely to have suffered "significant harm" or "very significant harm".

Mean scores for each barrier at each delay are shown in Table 2. The delay 1 barrier, *"the perceived financial costs associated with seeking care are too great"*, had the lowest mean combined score (i.e. considered most important). The *"lack of timely affordable emergency transport (formal or informal)"* was the barrier with the lowest mean combined score of importance within delay 2 (i.e. most important). Within delay 3, the *"lack of reliably available necessary physical resources (infrastructure, equipment and consumable material)"* had the lowest combined mean score followed by the *"lack of reliably available, suitably trained and motivated clinical staff"* (i.e. the most and second most important).

Across all 3 delays, the barrier most frequently reported within the top 3 most important was the delay 3 barrier, *"lack of reliably available necessary physical resources"* reported by

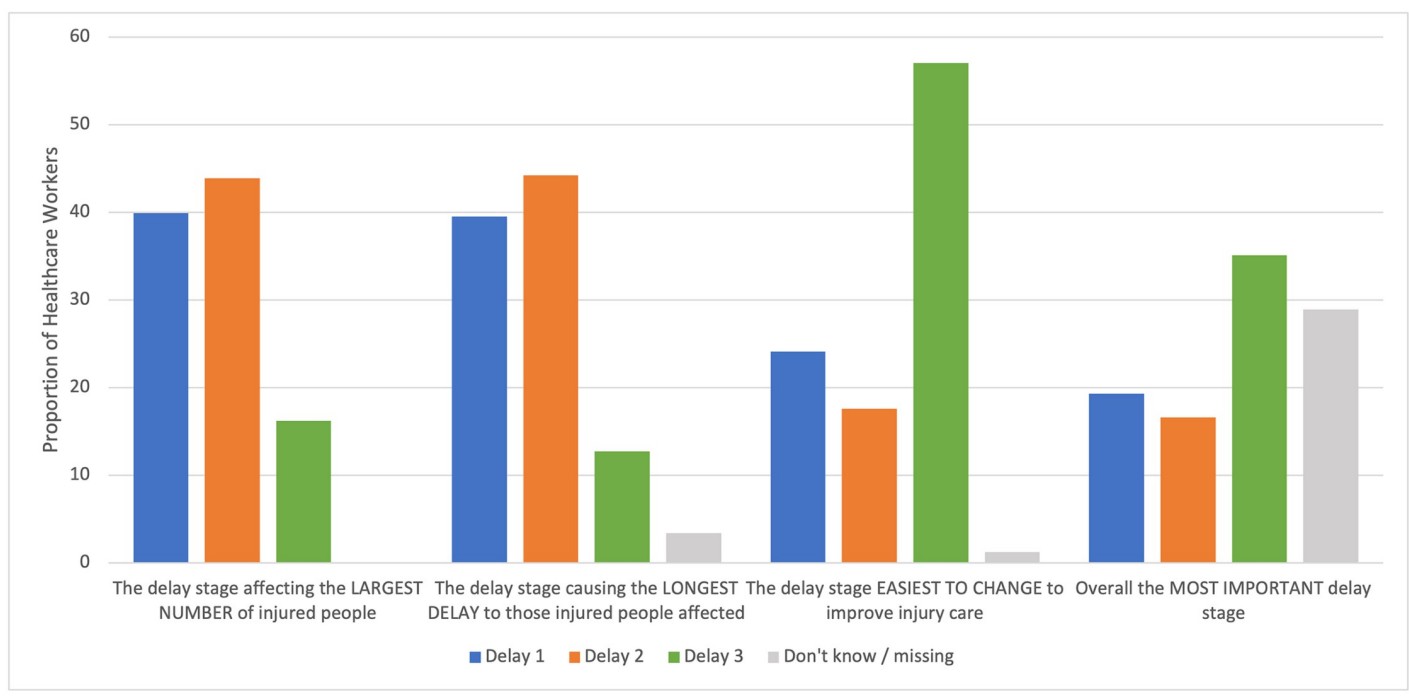

**Fig 1. Health facility staff perceptions of importance for each of the Three Delays.** Whilst delay 3 was not considered by most participants to either affect the largest number of people, nor cause the longest delay, most participants considered it the most amenable to change and the most important delay overall.

41.7% (95/228) participants (Fig 6). This was followed by the delay 3 barrier, *"lack of reliably available, suitably trained and motivated clinical staff"* by 39.0% (89/228) of participants. In third was the delay 2 barrier, *"lack of timely affordable emergency transport (formal or informal)"* by 29.8% (68/228) of participants.

Of the 27 additional barriers proposed by participants along with free text explanation, only one was deemed novel (S3 Table in S2 File). This novel barrier described occasions where patients are left unattended until a police report is obtained and proposed by the participant within delay 3 ("People are sent back to the police to collect a statement of injury").

## Discussion

We conducted a survey to understand how health facility staff perceive the most important barriers to high quality injury care within a Three Delays framework. Delay 3 was most commonly considered the most important delay although a substantial number of respondents did not commit to an answer. We found barriers within delay 3 were more commonly reported as being the most important with a lack of physical resources the most common. Financial costs and lack of transport were respectively identified as important barriers from the first and second delays.

Health facility staff are involved in caring for injured patients, most at least weekly in our study, as well as being members of the communities served by the facilities they work in. They are therefore well positioned to understand the issues affecting the systems they work within. Indeed, facility staff are commonly used as sources of data in studies on injury care in LMICs [10].

Whilst delay 3 was not considered by most participants to either affect the largest number of people, nor cause the longest delay, most participants considered it the most amenable to

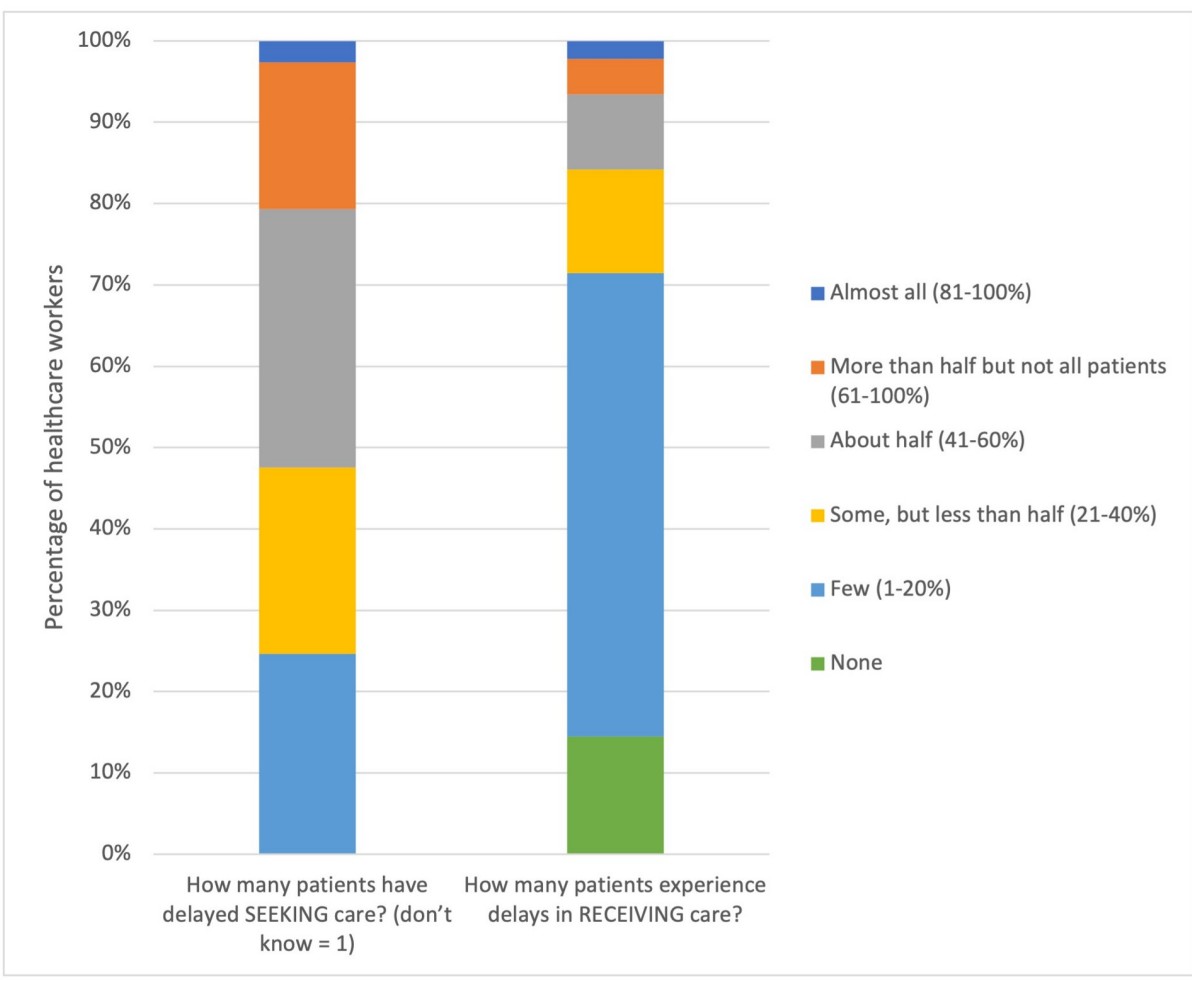

**Fig 2. Health facility staff perceptions of the proportion of patients delaying seeking and receiving care.** Most participants thought half or more patients delayed seeking care, but few or no patients experienced delays in receiving care.

change and the most important delay overall. This could be partly explained by staff believing more investment in physical and human resource, the priority barriers overall in our study, offers an expeditious way to meaningfully improve the system of care they work within [26]. Such investment would also likely tangibly benefit their own working conditions and wellbeing [27].

We asked participants to prioritise barriers both within each of the 3 delays and overall using two different approaches. Whilst the agreement between the findings was not perfect, the most important barriers within each delay were consistently prioritised through both approaches. This provides a means of internal validation and confidence that the findings represent the health facility staff priorities [28].

In Malawi, as is often the case in sub-Saharan Africa and other LMICs, facilities have been shown to lack resources compared to specified standards, most commonly WHO Essential Trauma Care [29–32]. A lack of infrastructure and physical resources inhibiting optimal care delivery to injured patients, especially in more peripheral locations, is evidenced across multiple study methods [33–38]. This is often markedly so at primary facilities, which, whilst not

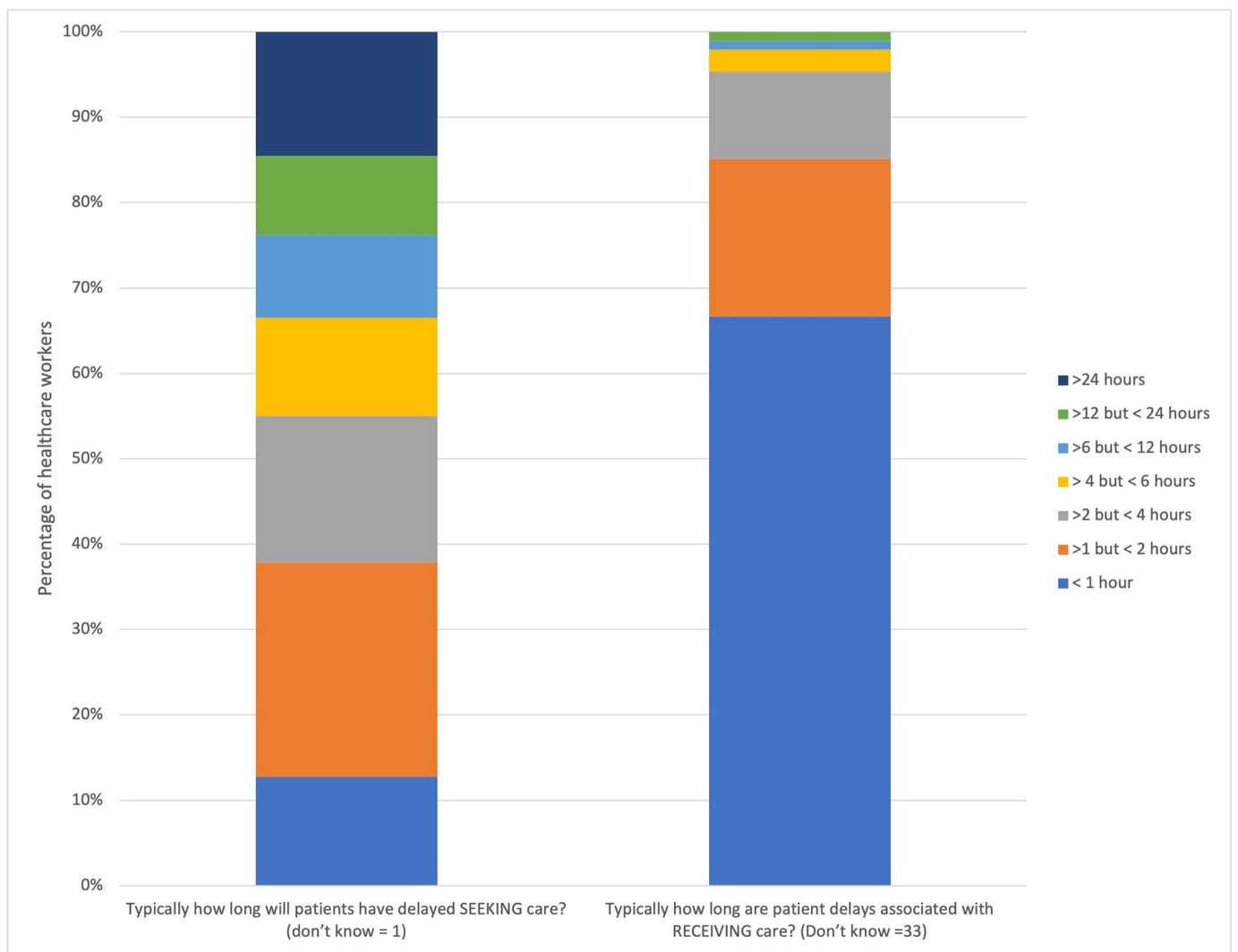

**Fig 3. Health facility staff perceptions of time delays to seeking and receiving care.** Most participants thought delays in patients seeking care were typically 2 hours or longer, whereas most thought delays in receiving care were less than 1 hour.

intended to offer specialist definitive curative care, are often the first location to initiate care for the injured [23, 29].

Economic issues of money and costs of seeking care were clearly perceived as the most important barrier within delay 1, seeking care. Cost implications intersect with other barriers within the Three Delays framework. Emergency transport costs, whilst largely informal, are especially pronounced in relation to fuel for motor vehicles and the barrier *"transport"*. Fears regarding potential liability for incurred costs, or losing income helping others contribute to the barrier *"community or bystander engagement"*. This can be both as a possible source of dispute or alternatively an inclination to prioritise one's own interests before assisting others. Patients may also preferentially seek *"traditional healers"* to avoid direct and indirect costs of formal health care [39].

Whilst the private or CHAM faith-based facilities in Malawi require direct payments for care [40, 41], only a minority of injured patients receive care at such facilities. Government

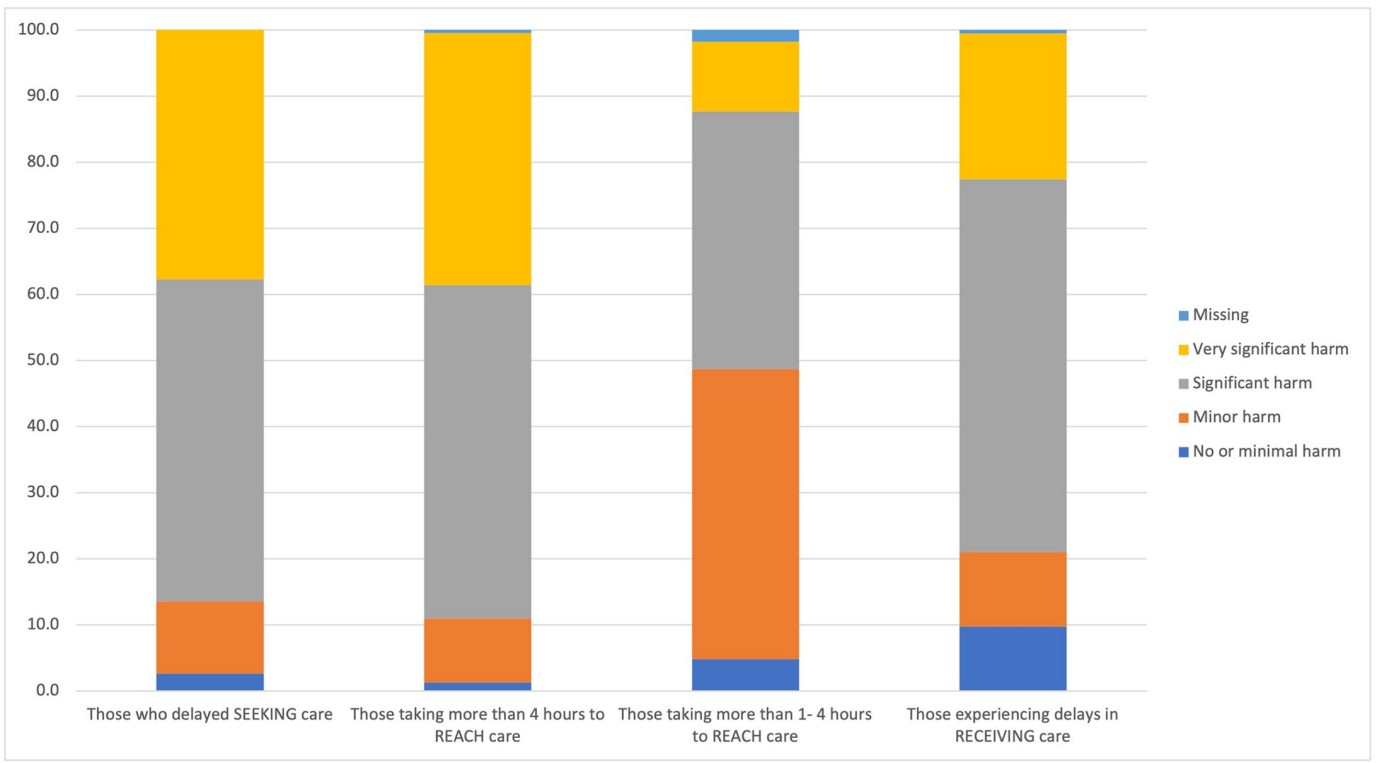

**Fig 4. Health facility staff perceptions of patient harm associated with each delay.** A substantial majority of participants considered patients with delays to seeking care, delays to reaching care of more than 4 hours and delays in receiving care suffered significant or very significant harm.

funded facilities provide the majority of care for those Malawians seeking care following injury. Such care should be free at point of access. Financial issues delaying or preventing care access have been found in comparable settings with community studies finding financial reasons for not seeking care between one tenth and a half of cases [39, 42–46].

Injuries can trigger substantial out of pocket expenditure [47, 48], including direct non-medical costs like travel and subsistence, as well as indirect effects related to lost income [49]. For example, in Burkina Faso seeking injury care was associated with needing to buy or sell something to meet these precipitant costs [50]. In the face of such financial hardship, a reluctance to seek care is understandable. Financial barriers inhibit timely urgent care seeking in LMICs for other conditions too [17, 51], a phenomenon also described in high income settings [52].

The funded nature of direct medical care costs in government facilities probably explains why problems with payment for care were not perceived as an important barrier in this study. In other settings such as Uganda, the payment of medical bills can be a common issue driving delay 3 [38]. Direct out of pocket payments needed for urgent surgery can be problematic for patients [53] and led to many discharging against medical advice in Nigeria [54]. Indeed, in some settings guarantees of payment are required in advance of treatment for injured patients [35].

Alleged corruption was not evidenced in this study of health facility staff. However, community and facility staff can differ in reporting alleged issues of client differentiation according to personal connections or finances in some less resourced systems [53]. Whilst informal payments can occur in developing economies, health facility staff insights on this subject probably

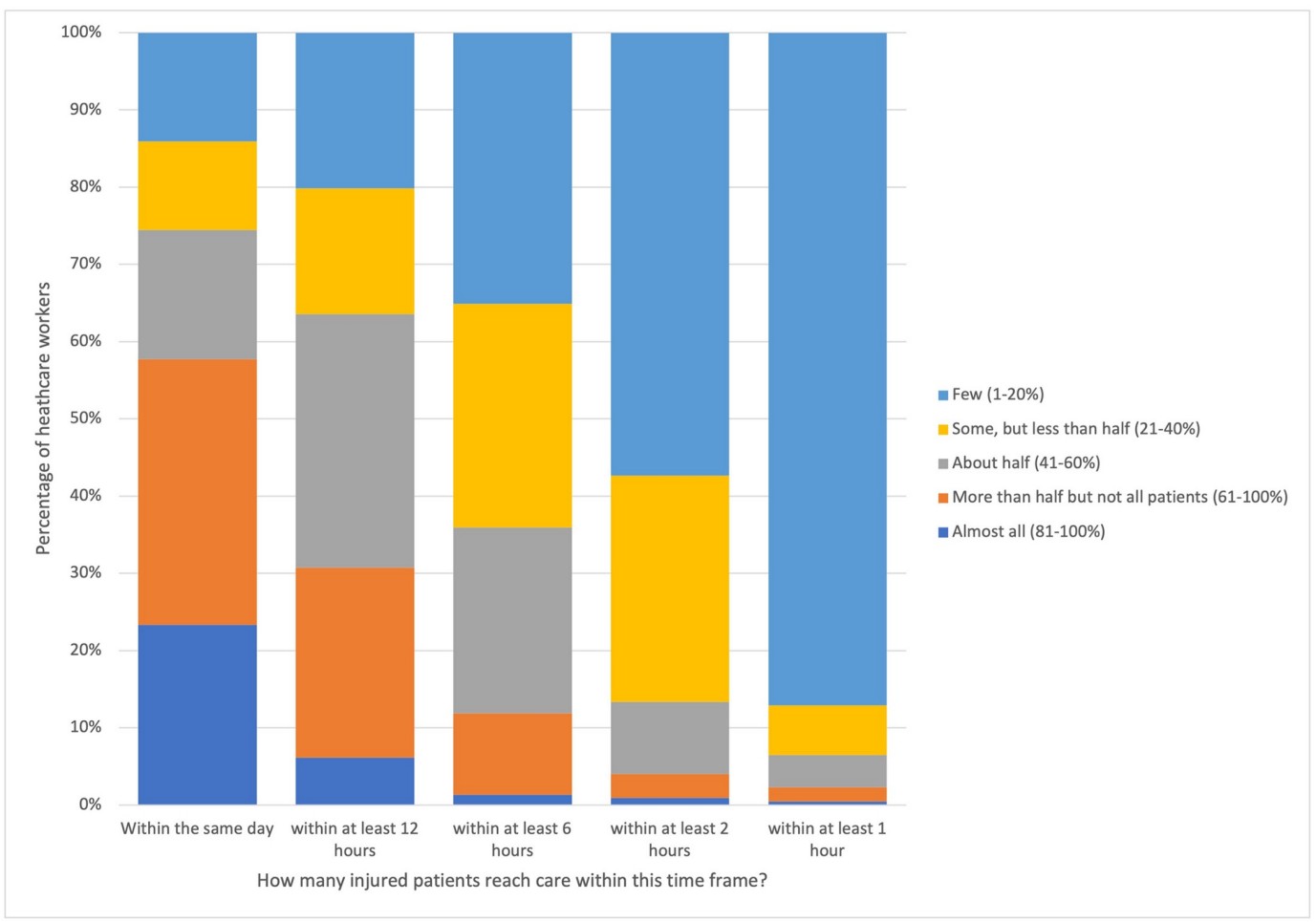

**Fig 5. Health facility staff perceptions of the proportion of injured patients reaching care within given time frames from injury.** The majority of participants thought few injured patients could reach care within 2 hours.

require specifically directed discussions, rather than more open approaches such as we used [53, 55]. Since legal or disciplinary implications could follow, a reluctance to discuss such issues is understandable [56]. Qualitative methods to investigate this complex and sensitive subject are potentially more suitable [56].

Our study found a lack of timely affordable emergency transport was the most important reported barrier within delay 2. Availability of professional prehospital care in Malawi is *"almost non-existent"* [57]. Where ambulances are found they are primarily for inter-facility transfer rather than prehospital use [57]. The situation is similar across much of sub-Saharan Africa as over 90% of the population lacks access to emergency medical services [58]. Informal methods are common in this and similar settings. Private vehicles and taxis are frequently used for transporting the injured instead of ambulances [33, 44, 46, 59–61]. There is evidence that interventions to improve prehospital trauma care in LMICs reduce injury related mortality [62]. However, where formal ambulance prehospital systems are present but immature, informal transport can still lead to more timely access to care than ambulances with limited availability [60, 61].

**Table 2. Heat map of barriers to care in order of mean combined importance score within each conceptual delay.**

| | Affecting the LARGEST NUMBER—mean score | Causing the LONGEST AMOUNT of delay—mean score | Mean combined score |
|---|---|---|---|
| **Delay 1 Seeking care**—Highest possible mean score 8 (16 combined) | | | |
| The perceived financial costs associated with seeking care are too great | 3.02 | 3.19 | 6.21 |
| Normal cultural behaviours delay seeking care such as gender roles, family responsibilities and requiring someone else's permission to seek care | 4.44 | 4.47 | 8.91 |
| People perceive that care is too difficult to physically access | 4.55 | 4.42 | 8.96 |
| People don't understand about health or available healthcare. | 4.28 | 4.82 | 9.08 |
| People perceive that available facility care is poor quality | 4.69 | 4.52 | 9.2 |
| There are delays in discovering injured people, including because of intoxication | 4.68 | 4.59 | 9.27 |
| People prefer traditional healers | 5.18 | 4.47 | 9.63 |
| People fear the consequences of helping an injured person, e.g. being accused of causing the injury. | 5.17 | 5.54 | 10.71 |
| **Delay 2 Reaching Care**—Highest possible mean score 6 (12 combined) | | | |
| There is a lack of timely affordable emergency transport (formal or informal) | 2.47 | 2.86 | 5.33 |
| There is a large physical distance from place of injury to an appropriate healthcare facility | 2.91 | 2.71 | 5.62 |
| There is a lack of timely available prehospital emergency care (formal or informal/bystander) | 3.4 | 3.54 | 6.93 |
| There is a lack of accessible emergency assistance communication mechanism (e.g. emergency call centre) | 3.21 | 3.74 | 6.95 |
| There is a lack of emergency care service coordination, including bypassing unsuitable facilities or transferring between facilities | 4.15 | 3.73 | 7.89 |
| There is a lack of reliable uncongested roads with priority for emergency vehicles | 4.86 | 4.42 | 9.28 |
| **Delay 3 Receiving Care**—Highest possible mean score 9 (18 combined) | | | |
| There is a lack of reliably available necessary physical resources (e.g. infrastructure, equipment and consumable material) | 2.99 | 3.17 | 6.15 |
| In regard to staffing, there is a lack of reliably available, suitably trained and motivated clinical staff | 3.22 | 3.85 | 7.07 |
| Specialist services needed for some injuries are not available in this area. | 3.79 | 3.5 | 7.29 |
| There is a lack of good quality, structured care processes for injured patients. | 4.67 | 4.66 | 9.33 |
| Lack of available means to safely and quickly transfer injured patients on to a more specialist facility. | 4.91 | 4.93 | 9.81 |
| In regard to patient demand, there is insufficient facility capacity to meet patient demand (e.g. overcrowding) | 5 | 4.94 | 9.93 |
| There is a lack of patient and family cooperation with care processes | 5.39 | 5.01 | 10.4 |
| Difficulties with timely payment for care | 7 | 6.96 | 13.96 |
| Need for unauthorised payments or gifts to healthcare staff to receive best available treatment. (e.g. corruption) | 8.04 | 8 | 16.03 |

Barriers ordered by participants were assigned a score corresponding to the order they were placed. Barriers affecting the first largest number of people were scored 1, those affecting second largest number of people were scored 2 etc. A mean score across all participants was then calculated for each barrier for a) affecting the largest number of patients, b) causing the longest amount of delay, and c) a combined mean barrier score (a+b). Lower scores indicate more important barriers. Colour scale according to mean score. Lower (more important) scores are closer to green, greater scores (less important) are closer to red.

As an exploratory analysis, we considered whether staff in referral (secondary or tertiary) facilities prioritised different barriers from the primary facility staff. This is shown in S4 Table of S2 File and shows that both primary and referral facility staff listed similar barriers amongst the top 3 most important. Interestingly, whilst both physical resources and staffing

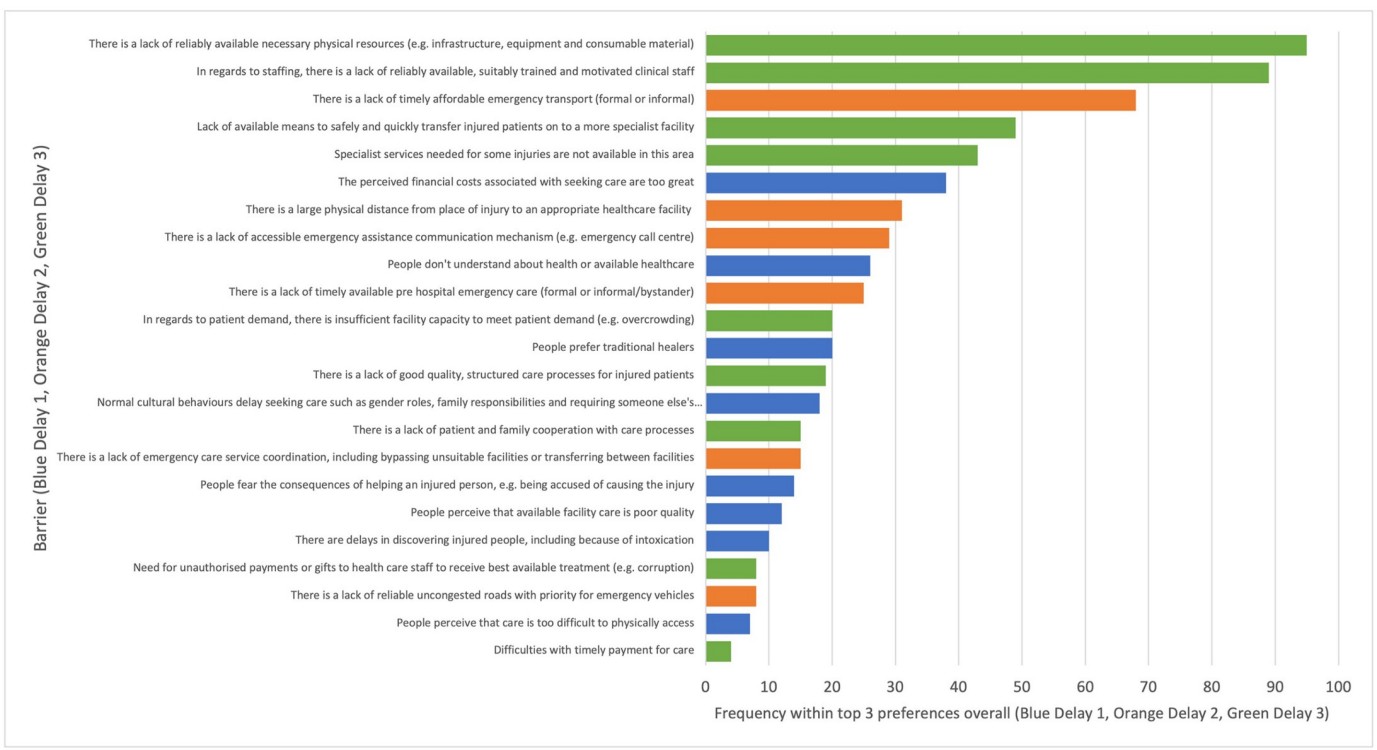

**Fig 6. Frequency of barriers reported by health facility staff within the top 3 most important overall (colour coded by delay stage).** Barriers from delay 3 were more frequently considered within the top 3 most important overall by participants.

were amongst the top 3 barriers for both referral and primary facility staff, referral facility staff more commonly prioritised physical resources whereas for in primary facilities staffing issue were more commonly prioritised, perhaps due to primary facility staff often working alone or in small numbers [8]. Furthermore, for primary facility participants, costs for seeking care and healthcare literacy were also relatively more often prioritised, perhaps related to their proximity to the communities where these issues are understood to prevent further care in referral facilities. Insufficient facility capacity was more commonly prioritised by referral facility staff, possibly related to a concentration of more unwell patients not able to be managed by primary facilities [63]. A lack of prehospital care was also relatively more often prioritised by referral facility staff, who perhaps more commonly experienced the consequences of patients who might have benefited from earlier therapeutic intervention on route to the facility [57].

There are limitations to our study to discuss. The survey was designed specifically for this study and its questions shaped within a Three Delays framework about barriers derived primarily from a Delphi study [14]. We surveyed a broad group of health facility staff including those who deliver care and services to the injured either directly as doctors and nurses or indirectly as other staff such as administrative, domestic or security staff [24]. This helped to garner a diverse range of perspectives on the problems and meant there was a range of educational attainment and potentially literacy amongst respondents. The survey introduced the Three Delays framework, likely for the first time for many, and may have been quite a complex concept for some to grasp. This was mitigated through careful piloting and training of administering the survey to ensure good understanding. However, the Three Delays framework can be limited as being barrier or problem focussed rather than prompting identification

or delivery of solutions [64]. In our study we have not explored the participants' ideas as to what should be done to address the barriers highlighted which could be an important avenue for further research.

The exposure and frequency that participants encountered injured patients varied, with at least annual involvement a minimum to be eligible. Requiring more frequent exposure might have provided more valid results but lessened the range of participants. However, almost two-thirds were involved in injury care at least weekly, whilst for over 90% this was at least monthly. The survey provided staff perceptions rather than objective patient-level data. Reliable patient-level data is not easy to come by in the absence of routine trauma registry maintenance [65, 66].

Participants may have focussed primarily on their facility environment. They may believe that facility-based care, for which they have at least some responsibility, is less of a problem than access related problems within other parts of the health system. This could perhaps explain the slightly conflicting survey finding that most participants believed delay 3 affected the least number of people and caused the least harm to those affected, but was nevertheless most often considered the most important. There was also a relative prioritisation seen of delay 3 barriers relative particularly to delay 1. This could be similarly explained by being foremost in the respondents' minds working in a facility context.

The Delphi derived barriers that framed the survey are broad categories which often encompass more than one concept [14]. This lessens the insight into which specific aspects of a barrier might be perceived as important. For example, within the barrier "*staff—in regards to staffing, there is a lack of reliably available, suitably trained and motivated clinical staff*", whether availability, training or motivation was perceived as most problematic has not been established. Further exploration would be required if that level of insight was desired. Additionally, the authors (TN and VK) conducting the surveys were both locally-based trained clinicians and as respected members of society may have elicited a form of social desirability amongst less qualified colleagues [67, 68]. This may have influenced respondents to present certain barriers, or even delay 3 as a whole, as problematic.

Our study was a relatively time-consuming method. This was partly a function of the number of participants, but also the nature of the questions. Asking opinions, perceptions and prioritising the relative importance of concepts can be unusual in the authors' experience delivering field research within this population. Electronic surveys are growing in use in health systems research [5], particularly given the availability of internet-connected smartphone devices [69]. However, face-to-face administered surveys tend to have a higher survey response rate and item completion rate, with the ability to more completely sample the target population through seeking out respondents of interest [70]. Only two authors (TN and VN) administered this survey, and additional staff would shorten the time for data collection. However, our survey could be done alongside other concurrent facility-based studies that might be included in a holistic multi method health system assessment.

Our study focused on health facility staff perspectives and did not have a patient voice. There are potential methodological, logistical, and ethical challenges to concurrently surveying patients during a time limited facility visit. For many of the primary facilities, there may not have been an injured patient present during the facility visit. At larger facilities, surveying patients during acute care has practical and ethical difficulties depending upon the nature of their injuries. We have previously used alternative methodological approaches such as community focus groups and household surveys to obtain a patient perspective complementary to the facility staff perspectives this study has captured [71, 72].

Whilst our study has demonstrated value in applying a Three Delays framed survey of health facility staff for injury health system evaluation, additional data sources and methods

are needed to more fully understand inherently complex and adaptive health systems [10]. Also, barriers to accessing quality care may differ between context and countries [73]. More rigorous evaluations using mixed methods approaches may be better suited to drive interventions and policy change [72, 73]. Future research through the NIHR funded Equi-Injury collaborative will advance a mixed methods Three Delays framed approach to injury care health system assessment in four different LMICs to co-develop improvements in equitable injury care [74].

## Conclusion

We surveyed health facility staff to establish their perceptions of the most important barriers to high quality injury care in their health system within a Three Delays format. Delay 3 and barriers within delay 3 were more commonly reported as being the most important with a lack of physical resources the most common. Financial costs and lack of transport were respectively identified as important barriers from the first and second delays. Health facility staff priorities should inform health system improvements, although specific findings may differ between health systems and from the priorities of community members and patients.

## Supporting information

**S1 File. STROBE checklist.**
(PDF)

**S2 File.** 1) List of barriers healthcare worker participants could select from, by delay, 2) Breakdown of other category of healthcare worker, 3) Other barriers proposed in the healthcare worker survey and their corresponding prioritisation, 4) Table demonstrating frequency of barriers reported by health facility staff within the top 3 most important overall according to participant place of work (primary or referral facility).
(PDF)

## Author Contributions

**Conceptualization:** John Whitaker, Abena S. Amoah, Rory Rickard, Andrew J. M. Leather, Justine Davies.

**Data curation:** John Whitaker, Taniel Njawala, Vitumbeku Nyirenda.

**Formal analysis:** John Whitaker.

**Funding acquisition:** John Whitaker, Rory Rickard, Andrew J. M. Leather, Justine Davies.

**Investigation:** John Whitaker, Taniel Njawala, Vitumbeku Nyirenda, Abena S. Amoah, Albert Dube.

**Methodology:** John Whitaker, Abena S. Amoah, Albert Dube, Andrew J. M. Leather, Justine Davies.

**Project administration:** John Whitaker, Taniel Njawala, Vitumbeku Nyirenda, Abena S. Amoah, Albert Dube, Lindani Chirwa, Boston Munthali.

**Resources:** John Whitaker, Taniel Njawala, Vitumbeku Nyirenda, Abena S. Amoah, Albert Dube, Lindani Chirwa, Boston Munthali, Justine Davies.

**Supervision:** John Whitaker, Rory Rickard, Andrew J. M. Leather, Justine Davies.

**Visualization:** John Whitaker.

**Writing – original draft:** John Whitaker.

**Writing – review & editing:** John Whitaker, Taniel Njawala, Vitumbeku Nyirenda, Abena S. Amoah, Albert Dube, Lindani Chirwa, Boston Munthali, Rory Rickard, Andrew J. M. Leather, Justine Davies.

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
