## [Decision Letter · Decision Letter 0]

13 Mar 2024

PONE-D-24-01288Identifying and prioritising barriers to injury care in Northern Malawi, results of a multifacility multidisciplinary health facility staff surveyPLOS ONE

Dear Dr. Whitaker,

Thank you for submitting your manuscript to PLOS ONE. After careful consideration, we feel that it has merit but does not fully meet PLOS ONE’s publication criteria as it currently stands. Therefore, we invite you to submit a revised version of the manuscript that addresses the points raised during the review process.

We look forward to receiving your revised manuscript.

Kind regards,

Novel N. Chegou, Ph.D

Academic Editor

PLOS ONE

Journal Requirements:

"This work was supported by a research fellowship awarded to JW from the Royal

College of Surgeons of England and The King’s Centre for Global Health and Health

Partnerships."

Reviewers' comments:

Reviewer's Responses to Questions

**Comments to the Author**

1. Is the manuscript technically sound, and do the data support the conclusions?

Reviewer #1: Yes

2. Has the statistical analysis been performed appropriately and rigorously? 

Reviewer #1: Yes

3. Have the authors made all data underlying the findings in their manuscript fully available?

Reviewer #1: Yes

4. Is the manuscript presented in an intelligible fashion and written in standard English?

Reviewer #1: Yes

5. Review Comments to the Author

Reviewer #1: The authors report on the results of a survey of health care workers, primarily, from several health care centers in Malawi. A total of 228 workers, mostly clinicians, were asked about barriers that lead to delays in either seeking, reaching, or receiving injury (trauma) care. Overall, the manuscript is very well written with minimal need for editing (see below single comment). However, the results are not surprising, given the population interviewed, and mirror those of other surveys from sub-Saharan Africa. As mentioned by the authors, there are key differences between centers that likely influence the prioritization given to the questions posed. For example, the faith-based facilities required payment for services, making this a much more significant barrier for patients, but could lead to better resource availability due to reduced demand and higher revenue. As a result of pooling all the responses together, the authors likely suppressed any key differences in barriers to care seeking and delivery that may exist between care delivery models. The analysis would be more informative if it was analyzed by site type. Public vs Military vs Private (faith-based). Or if that is not possible, at least by level, so that responses at the tertiary facility are looked at separate from the “front line” facilities.

While the perspectives of the care providers were argued to also reflect the views of the patients in the communities served, this is not necessarily true. Why weren’t patients surveyed? Were they not also in the facilities that were visited? Having both perspectives and looking at areas of alignment would have been more useful in planning and design of future interventions.

It is human nature to say that you could do a better job if you had more resources (even in HICs), so this too is a significant bias affecting the survey results. While it is simple to say the facilities lack human and supply resources and this is perceived as easily addressable, the reality of creating a sustainable change in any health care related resource is far from simple. Thus, it is of interest to know how these providers felt this could be easily achieved? What is it they needed that was easily implemented? Again, this is key to making the reported information useful in future efforts to improve care in the region. From the data presented in figure 6, the lack of transport to a facility for care stands out, with multiple orange bars essentially highlighting the cost and availability of transport as a major barrier. From the time data, about half seem to get care within 2 hours, but the remaining half face much longer delays, and this is perceived as being very harmful. Thus, the greatest impact on the greatest number would seem to be getting patients to the care that is available, and not necessarily changing the care that is delivered in under an hour for most, based on the data. This conclusion would seem to be at odds with the current conclusions of the manuscript. “Health facility staff priorities should inform health system improvements, although specific findings may differ between health systems and from the priorities of community members and patients.” One could argue, alternatively, in a resource limited environment, the greatest good for the greatest number should, perhaps, drive policy.

While some of the biases and influences on the results are part of the discussion throughout, the formal limitation paragraph is underdeveloped and should include a concise summary of the biases and limitations of the approach taken. What were limits on analysis? Why did you consolidate so many categories into dichotomous groups? Presumably there were some key limitations to the analyses that were the bases for this and should be addressed. If the above suggested analyses can’t be performed, then this should also be addressed.

As presented, the results and discussion do not provide a foundation or direction for future efforts to impact the care delivery in the region. Presumably the purpose for this work was to identify barriers that could be addressed. Thus, consolidating some of the discussion to focus on the key results and the implications for future efforts to impact care would be welcome changes in the manuscript.

One typo noted on page 14: I think you may want to change “meaningful” to “meaningfully”

“…offers an expeditious way to meaningful improve the system of care they work within…”

6. PLOS authors have the option to publish the peer review history of their article (what does this mean?). If published, this will include your full peer review and any attached files.

Reviewer #1: No

---

## [Author Response · Author response to Decision Letter 0]

30 Apr 2024

See uploaded response to reviewer document.

---

## [Decision Letter · Decision Letter 1]

15 Jul 2024

PONE-D-24-01288R1Identifying and prioritising barriers to injury care in Northern Malawi, results of a multifacility multidisciplinary health facility staff surveyPLOS ONE

Dear Dr. Whitaker,

Thank you for submitting your manuscript to PLOS ONE. After careful consideration, we feel that it has merit but does not fully meet PLOS ONE’s publication criteria as it currently stands. Therefore, we invite you to submit a revised version of the manuscript that addresses the points raised during the review process.

We look forward to receiving your revised manuscript.

Kind regards,

Novel N Chegou, Ph.D

Academic Editor

PLOS ONE

Journal Requirements:

Additional Editor Comments:

The reviewer has identified additional important concerns regarding the figures - color scheme and quality, and figure legends. Please attend to these and other issues pointed out by the reviewer.

Reviewers' comments:

Reviewer's Responses to Questions

**Comments to the Author**

1. If the authors have adequately addressed your comments raised in a previous round of review and you feel that this manuscript is now acceptable for publication, you may indicate that here to bypass the “Comments to the Author” section, enter your conflict of interest statement in the “Confidential to Editor” section, and submit your "Accept" recommendation.

Reviewer #1: All comments have been addressed

2. Is the manuscript technically sound, and do the data support the conclusions?

Reviewer #1: Yes

3. Has the statistical analysis been performed appropriately and rigorously? 

Reviewer #1: N/A

4. Have the authors made all data underlying the findings in their manuscript fully available?

Reviewer #1: Yes

5. Is the manuscript presented in an intelligible fashion and written in standard English?

Reviewer #1: Yes

6. Review Comments to the Author

**Reviewer #1:** The authors have addressed the primary concerns raised by the initial review. A few minor edits should be made prior to publication. Regarding the figures, Figures 1 and 6 should use the same colors to represent the given delays. It is noted that figure titles are provided within the text, presumably as place holders for figure location, but these should also have figure legends to support interpretation of the figure as a stand-alone presentation of the data. Key findings should be noted in the legend and any significant differences upon which statistical analysis was performed should be highlighted. This latter aspect may not be relevant for the figures in this manuscript, but may be relevant to any supplemental data and figures. Finally the image quality seems low, this is a consistent problem for the default graphic outputs of Microsoft office applications. Check to be sure exports of images are at least 600 DPI which will require some manual adjustment of these output settings.

7. PLOS authors have the option to publish the peer review history of their article (what does this mean?). If published, this will include your full peer review and any attached files.

Reviewer #1: No

---

## [Author Response · Author response to Decision Letter 1]

23 Jul 2024

Please see attached response to reviewer comments.

---

## [Editor Report · Decision Letter 2]

25 Jul 2024

Identifying and prioritising barriers to injury care in Northern Malawi, results of a multifacility multidisciplinary health facility staff survey

PONE-D-24-01288R2

Dear John

We’re pleased to inform you that your manuscript has been judged scientifically suitable for publication and will be formally accepted for publication once it meets all outstanding technical requirements.

Kind regards,

Novel N. Chegou, Ph.D

Academic Editor

PLOS ONE
---

## [Editor Report · Acceptance letter]

26 Jul 2024

PONE-D-24-01288R2 

PLOS ONE

Dear Dr. Whitaker, 

I'm pleased to inform you that your manuscript has been deemed suitable for publication in PLOS ONE. Congratulations! Your manuscript is now being handed over to our production team.

Kind regards, 

on behalf of

Prof Novel Njweipi Chegou 

Academic Editor

PLOS ONE